# Temperature Response of Metabolic Activity of an Antarctic Nematode

**DOI:** 10.3390/biology12010109

**Published:** 2023-01-10

**Authors:** Colin Michael Robinson, Lee D. Hansen, Xia Xue, Byron J. Adams

**Affiliations:** 1Department of Biology, Brigham Young University, Provo, UT 84602, USA; 2Department of Chemistry and Biochemistry, Brigham Young University, Provo, UT 84602, USA; 3Henan Key Laboratory of *Helicobacter pylori* & Microbiota and Gastrointestinal Cancer, Marshall Medical Research Center, The Fifth Affiliated Hospital of Zhengzhou, Zhengzhou University, Zhengzhou 450000, China; 4Monte L. Bean Life Science Museum, Provo, UT 84602, USA

**Keywords:** Antarctica, carbon cycling, climate change, nematode, respiration rates, soil temperature

## Abstract

**Simple Summary:**

To understand how the McMurdo Dry Valleys of Antarctica (MCM) will respond to climate change, it is necessary to understand how dominant organisms in the ecosystem respond to fluctuations in temperature and water availability. We studied the effect of temperature on the metabolic activity of *Plectus murrayi,* a widespread nematode in the MCM. By analyzing heat produced by metabolism along with CO_2_ production and O_2_ consumption, we found *P. murrayi* reaches peak metabolic activity at 40 °C, an unexpectedly high metabolic threshold for an Antarctic organism. As temperatures rise in the MCM, so too will the metabolic activity of *P. murrayi.* Such increases in energy demands have the potential to disrupt soil ecosystem structure and functioning, as the MCM system is carbon limited. Should *P. murrayi* experience heightened metabolic activity for extended periods of time, without additional carbon inputs the functioning of these soil ecosystems in the MCM may become significantly reduced.

**Abstract:**

Because of climate change, the McMurdo Dry Valleys of Antarctica (MCM) have experienced an increase in the frequency and magnitude of summer pulse warming and surface ice and snow melting events. In response to these environmental changes, some nematode species in the MCM have experienced steady population declines over the last three decades, but *Plectus murrayi*, a mesophilic nematode species, has responded with a steady increase in range and abundance. To determine how *P. murrayi* responds to increasing temperatures, we measured metabolic heat and CO_2_ production rates and calculated O_2_ consumption rates as a function of temperature at 5 °C intervals from 5 to 50 °C. Heat, CO_2_ production, and O_2_ consumption rates increase approximately exponentially up to 40 °C, a temperature never experienced in their polar habitat. Metabolic rates decline rapidly above 40 °C and are irreversibly lost at 50 °C due to thermal stress and mortality. *Caenorhabditis elegans*, a much more widespread nematode that is found in more temperate environments reaches peak metabolic heat rate at just 27 °C, above which it experiences high mortality due to thermal stress. At temperatures from 10 to 40 °C, *P. murrayi* produces about 6 times more CO_2_ than the O_2_ it consumes, a respiratory quotient indicative of either acetogenesis or de novo lipogenesis. No potential acetogenic microbes were identified in the *P. murrayi* microbiome, suggesting that *P. murrayi* is producing increased CO_2_ as a byproduct of de novo lipogenesis. This phenomenon, in conjunction with increased summer temperatures in their polar habitat, will likely lead to increased demand for carbon and subsequent increases in CO_2_ production, population abundance, and range expansion. If such changes are not concomitant with increased carbon inputs, we predict the MCM soil ecosystems will experience dramatic declines in functional and taxonomic diversity.

## 1. Introduction

Climate changes are occurring worldwide but are predicted to occur faster and at a higher magnitude in the Polar regions [1]. The most rapid increases in temperature have occurred in Antarctica, where average annual temperature has increased by almost 3 °C over the past 50 years [2]. Over the next century, Antarctic temperatures are expected to rise at an even higher rate, resulting in lengthening melting seasons and an increase in precipitation [3,4].

The McMurdo Dry Valleys (MCM) of Antarctica, which have experienced relatively modest warming compared to the rest of the continent, are part of the coldest and driest desert on the planet [5]. As a result of climate change, the MCM have experienced gradually warming summers and more frequent heat waves since 2001 [6]. Nematodes are the most abundant and widely distributed metazoans in the MCM [7]. The four main nematode taxa in the MCM are *Scottnema*, *Eudorylaimus*, *Geomonhystera*, and *Plectus* [8]. These nematodes are well adapted to the cold temperatures and extreme desiccation of the MCM [9] and slight changes in water availability and temperature can have large impacts on nematode communities in this region [10,11,12].

Populations of *Scottnema*, which thrive in cold, dry, salty soil habitats, have been declining for the past three decades in response to increased soil moisture and temperature [6,13,14]. On the other hand, *Plectus murrayi*, a less common nematode of MCM landscapes, inhabits soils that are less harsh, typically wetter, and less salty [15,16], and has experienced consistent population growth and range expansion since 2001 [6,17,18]. Populations of *P. murrayi* have also been seen to increase in passive greenhouses where temperature and moisture levels are higher than in the natural environment [19].

Because the MCM are home to some of the most organic-poor soils on the planet with organic carbon consistently below 0.1 wt% [20,21], population expansion of any nematode species will likely have lasting impacts on carbon cycling and soil community composition. This is especially the case if expanding species are also experiencing increased metabolic activity due to increasing temperatures. Many species, including house fly pupae (*Musca domestica*), third instar ladybugs (*Harmonia axyridis*), and fifth instar codling moths (*Cydia pomonella*), have been shown to experience heightened metabolic heat rate at elevated temperatures [22,23,24]. In most cases, however, ectotherms reach peak metabolic heat rate at a temperature reflected by their natural habitat. To determine whether *P. murrayi* could experience heightened metabolic activity as a result of warming climate, we measured the metabolic response of *P. murrayi* to increasing temperatures. As *P. murrayi* is the only nematode from the MCM to date which can be cultured, we were unable to compare its metabolic response to any other nematode from the same habitat. Thus, for comparative purposes, we chose another free living soil microbivore, the well-studied N2 strain of *C. elegans*. Because soil temperatures in the MCM experience an annual mean temperature of −26.1 °C with an absolute minimum of −58.2 °C and an absolute maximum of 22.7 °C and *P. murrayi* seems to be responding positively as temperatures rise to near the absolute maximum for longer periods of time, we predicted that *P. murrayi* would likely reach peak metabolic capacity near 22.7 °C [5].

Measurements of *P. murrayi* metabolic rates inform predictions about how *P. murrayi* will respond to future climate changes. By furthering our understanding of how the metabolic response of *P. murrayi* will react to future climate changes, we can better predict future effects of *P. murrayi* metabolism on carbon cycling, patterns of nematode species abundance and distribution in the MCM, and contributions of these effects to local, regional, and global CO_2_ production. In this way, an understanding of how individual nematode populations will respond to climate change can help us better understand the unique soil ecosystems in the MCM and how climate change might impact them in the future.

## 2. Materials and Methods

### 2.1. Nematode Isolation

Soil samples were collected from Taylor Valley, Antarctica. Soil cores to 10 cm depth were removed using clean plastic scoops, placed in sterile Whirlpak^®^ bags, and transported in insulated coolers via helicopter to McMurdo Station. The soil samples were gradually cooled to −20 °C (at a rate of −10 °C per 48 h) and shipped frozen to Brigham Young University. Soils were then gradually warmed to +4 °C (at a rate of +10 °C per 48 h). Nematodes were extracted from the soil using sugar density gradient centrifugation modified for Antarctic soils [25,26]. *P. murrayi* were isolated and cultures established according to Adhikari and Tomasel et al. (2010). Cultured *P. murrayi* were then placed in deionized water and stored at −20 °C.

Agar liquid media was then prepared with double deionized water at a concentration of 15 g/L. Fifteen grams of agar powder (Thermo Fisher Scientific, Ward Hill, MA, USA) was stirred into 965 mL of double deionized water until a homogenous translucent liquid formed. Twenty milliliters of Bold’s modified basal media (Sigma-Aldrich, St. Louis, MO, USA) was added and pH was adjusted to 7 with 0.1 M NaOH and 0.1 M HCl. The solution was then made up to 1 L with double deionized water. The liquid media was then autoclaved with a 20 min sterilization step at 120 °C and then poured into 60 mm petri dishes until they were approximately 2/3 full. Before agar was allowed to set, 2 g of sterilized Standard Ottawa Sand (EMD Chemical, Gibbstown, NJ, USA) was added to the center of each plate due to the observed improved viability of nematodes in the presence of sand. Sealed plates were held at room temperature for 3 days to monitor contamination.

Uncontaminated plates were then prepared with 40 µL of pure OP50 *Escherichia coli* culture that had been tested for contaminants and incubated at 37 °C for 3 days. *P. murrayi* isolates which had been stored at −20 °C were then thawed and deposited on *E. coli* plates and held at 11 °C for a 4-week population expansion period. The living cultures were maintained by preparing additional agar plates with *E. coli* and using a sterile knife to transfer pieces of agar containing live nematodes from the old plates to the new ones. Agar transfers were carried out every 4 weeks to maintain the viability and health of the worms and to provide them with fresh *E. coli*.

### 2.2. Microcalorimetric Measurements of Heat and CO_2_ Production Rates

A TAM IV isothermal microcalorimeter (TA Instruments, Lindon, UT) was used to measure metabolic heat and CO_2_ production rates via calorespirometry. Six pieces of agar populated with a 2-week-old living culture of *P. murrayi* were excised using a sterile knife; the nematodes upon them were counted under a dissection microscope (27–56 nematodes); and they were each placed in one of six 4 mL vials. A 250 μL ampoule was then added to each of the six 4 mL vials: an ampoule containing 200 µL of 0.4 M NaOH in three vials, an ampoule with 200 µL of 0.4 M NaCl in two vials, and an ampoule with 200 µL of ddH2O in one vial [22,27]. The six 4 mL vials were then sealed and inserted into the six-channel calorimeter in the TAM IV (Figure 1).

During the experiment, CO_2_ produced by metabolism reacts with the NaOH in the 4 mL vials to produce sodium carbonate and water, releasing 108.5 kJ of heat per mole of CO_2_. The difference in measured heat rate between the vials with NaOH and the vials without NaOH divided by 108.5 kJ/mole CO_2_ thus provides the rate of CO_2_ production [28,29]. The heat produced per mole CO_2_ was assumed to be independent of temperature from 5 to 50 °C. O_2_ consumption was calculated from the measured heat rates from the vials without NaOH with Thornton’s Rule; 455 kJ of heat is produced per mole O_2_ consumed [30].

The TAM IV was programmed to measure heat produced per vial at each of the following temperatures sequentially: 15 °C, 10 °C, 5 °C, 15 °C, 20 °C, 25 °C, 30 °C, 35 °C, 40 °C, 45 °C, 50 °C, and then back to 15 °C. The 5-degree transitions between temperatures took approximately 1.5 h each. Vials were held at each temperature for 4 h, during which time heat rate measurements were recorded every 5 s. After a thermal equilibration period of about 30 min at each temperature, the measurements were averaged for each vial at each temperature setting. This experiment was then repeated in 5 vials containing a known number of *C. elegans* (20–31) and no NaOH. Baseline values for the heat rate measurements were obtained with a vial containing only agar, 200 µL of 0.4 M NaOH, and the same amount of OP50 *E. coli* as was used in the nematode experiments. Baseline heat rates were all 0 ± 1 μJ/s at all temperatures with *E. coli* producing a negligible amount of heat, so no baseline correction was done.

The average heat rate for each vial with *P. murrayi* or *C. elegans* was then divided by the number of nematodes in the respective vial to calculate the average heat produced per nematode at each temperature. *P. murrayi* with NaOH was then compared to *P. murrayi* without NaOH using a mixed ANOVA test with a between-subjects variable of treatment and a within-subjects variable of temperature (N = 6 vials). *P. murrayi* without NaOH was then compared to *C. elegans* using a mixed ANOVA test with a between-subjects variable of species and a within-subjects variable of temperature (N = 8 vials). Pairwise comparison analysis was conducted at each temperature point using a paired T-test which was corrected for multiple measures using the Bonferroni correction. The average heat produced per *P. murrayi* individual at each temperature was then used to calculate the average rate of O_2_ production per nematode. The averaged heat rate per *P. murrayi* nematode in the vials without NaOH was then subtracted from the averaged heat rate per nematode from the vials with NaOH to obtain the average heat rate from CO_2_ reacting with NaOH at each temperature. This value was then used to calculate the average CO_2_ production rate per nematode at each temperature. The moles of CO_2_ produced per second per nematode was then divided by the moles of O_2_ consumed per second per nematode to calculate the respiratory quotient of *P. murrayi* metabolism.

## 3. Results

Total heat rates from samples containing *P. murrayi* varied from about 4 μJ/s at 5 °C to about 30 μJ/s at 40 °C (Figure A1). The average heat rates per *P. murrayi* individual in vials with and without NaOH are shown in Figure 2A. All vials with *P. murrayi* showed increasing heat rates as the temperature increased from 5 °C to 40 °C. The peak heat rate is at 40 °C in both curves, 1.49 ± 0.08 μJ per second per nematode in the presence of 0.4 M NaOH and 0.66 ± 0.08 μJ per second per nematode in the absence of NaOH. There was a statistically significant interaction between *P. murrayi* treatment group and temperature in explaining the heat rate. Pairwise comparisons show that the mean heat rate was significantly different between the two *P. murrayi* groups at all temperatures except 45 °C and 50 °C. The difference in heat rate is from the exothermic reaction of NaOH with CO_2_ produced by nematode metabolism. At temperatures above 40 °C, a sharp decline in heat rate occurred. After being held at 50 °C for 4 h and then returned to 15 °C, all vials containing *P. murrayi* registered an average heat rate of 0 ± 1 μW, indicating the nematodes had died from thermal stress. All vials with *C. elegans* showed increasing heat rates as the temperature increased from 5 °C to 25 °C. At temperatures above 25 °C, vials with *C. elegans* showed a sharp decline in heat rate, with all vials reaching heat rates of 0 ± 1 μJ/s between 30 and 45 °C (Figure A2).

The CO_2_ production rate and O_2_ consumption rate per *P. murrayi* nematode are shown in Figure 3A. The molar ratio of CO_2_ produced to O_2_ consumed by *P. murrayi*, otherwise known as the respiratory quotient, was highest at 10 and 15 °C with a value of 6.8 ± 2 moles of CO_2_ produced per mole of O_2_ consumed at both temperatures (Figure 3B). The ratio is minimal at 5 °C, 3.6 ± 1.5, and trends downward as temperature increases from 15 to 50 °C. The fraction of CO_2_ produced by oxidative respiration at each temperature follows an inverse trend to the respiratory quotient, reaching its lowest point at 15 °C with a ratio of 0.12 ± 0.03 (Figure 3C).

## 4. Discussion

Because *P. murrayi* releases the most heat, produces the most CO_2_, and consumes the most O_2_ at 40 °C, we conclude that *P. murrayi* is reaching peak metabolic activity at that temperature. This is similar to the temperature response measured in house fly (*Musca domestica*) pupae, third instar ladybugs (*Harmonia axyridis*), and fifth instar codling moths (*Cydia pomonella*), which reach peak metabolic activity at 41 °C, 38 °C, and 40 °C, respectively [22,23,24]. However, an Antarctic ectotherm, *P. murrayi*, reaching peak metabolic activity at 40 °C is unprecedented and surprising. This may be a relic of an evolutionary past in which the ancestors of *P. murrayi* experienced higher temperatures than they do now in the MCM, or a metabolic anomaly. As we were unable to compare this response to other species in a phylogenetic context, the evolutionary origin and maintenance of the metabolic response remains speculative. Furthermore, seeing as this study was conducted under synthetic conditions, further work should focus on determining whether *P. murrayi* exhibits increased grazing on soil microbes under natural warm conditions.

Though there is no direct connection to be made between heightened metabolic activity and population expansion, it may be a protective exaptation allowing for the recent success of *P. murrayi* nematodes during the present period of warming. Although *P. murrayi* showed peak metabolic activity at 40 °C, preliminary studies show that *P. murrayi* only experiences a few days of heightened physical and reproductive activity at temperatures above 30 °C before experiencing high rates of mortality due to a heightened rate of living. It is unlikely, however, for temperatures in the MCM to exceed 30 °C soon and certainly not for extended periods of time. Therefore, if current climate trends in the MCM continue, we can expect *P. murrayi* to continue to function at a heightened metabolic capacity for longer periods of time.

### Ecological Impacts

An increase in *P. murrayi* metabolic activity could have significant impacts on carbon cycling in the MCM. Nematode communities contribute between 2% and 7% of heterotrophic carbon flux in the Dry Valleys [31]. Decreased abundance of *S. lindsayae* has already been seen to significantly reduce carbon cycling in the MCM, leading to a decrease in soil carbon depletion rates [31]. An increase in *P. murrayi* metabolic activity could increase carbon cycling as they pull more organic carbon out of the soil. In the MCM, which is one of the most organically deplete systems on Earth, available soil organic carbon is a limiting nutrient for suitable nematode habitat [20,21]. Without a concomitant increase in contemporary carbon inputs, metabolic activity of *P. murrayi* may ultimately be controlled by carbon depletion. However, recent research shows that Antarctic phototrophs can increase carbon fixation under warmer temperature regimes, suggesting that available soil organic carbon may become less limiting as a result of climate change [32].

Our findings also suggest that *P. murrayi* releases far more consumed carbon in the form of CO_2_ than would be expected by typical metabolic activity. Between 5 and 15 °C, the ratio of CO_2_ produced to O_2_ consumed nearly doubles, rising from ~3.88 to ~6.85 moles of CO_2_ produced for every 1 mole of O_2_ consumed. The digestion of carbohydrates produces ~0.8 moles of CO_2_ for every 1 mole of O_2_ consumed. The unusually high ratio of CO_2_ production to O_2_ consumption in *P. murrayi* indicates that more carbon is being pulled from the soil then would be expected under typical respiration and must therefore indicate an atypical metabolic pathway.

On average, only 18% of the CO_2_ produced per nematode is accounted for by oxidation of ingested carbohydrates by O_2_. Because the ingested food, *E. coli*, has close to the same average oxidation state of carbon as the nematode biomass, the remaining 82% of CO_2_ produced cannot result from oxidation reactions. One potential explanation for the excess CO_2_ is that it is produced by a symbiotic acetogenic microbe through the reaction of acids produced during metabolism with HCO_3_^−^ ingested along with the food [33]. However, although HCO_3_^−^ is plentiful in the native Antarctic soils where the nematodes originated [34], it is not present in the gelled media used in this study. Furthermore, we did not find significant alignments for any acetogenic microbes in the *P. murrayi* microbiome (Accession number: SAMN19844092).

As we found no other reasonable source for the excess CO_2_ in contemporary literature, we conclude that it is likely produced as sugars obtained from the agar media are converted to lipids via de novo lipogenesis. Such metabolic pathways have been characterized in *C. elegans* where glucose molecules undergo partial glycolysis to form two molecules of acetyl-CoA which then combine to begin formation of a fatty acid chain, releasing one molecule of CO_2_ [35,36]. Increased de novo lipogenesis has also been shown to elevate the CO_2_ production and respiratory quotient of locusts [37].

The ratio of CO_2_ produced by respiration to CO_2_ produced by lipogenesis fluctuates with temperature. At 5 °C, 22% of CO_2_ produced by *P. murrayi* is accounted for by respiration. At 10 °C, respiratory CO_2_ drops to just 12% of the total CO_2_ produced. This indicates that elevated temperatures and heat shock lead to an increase in lipid production in *P. murrayi*. Most, if not all of these lipids are likely being produced to store energy in anticipation of coming stress. However, many poikilotherms adjust the saturation of lipid membranes in response to temperature, increasing saturation in response to heat shock and decreasing saturation in response to cold shock [38]. Therefore, *P. murrayi* may also experience an increase in saturated lipid production at elevated temperatures to maintain cell membrane integrity.

Although *P. murrayi* does not often experience heat shock in its native environment, it does have active heat shock proteins that it uses in response to extreme freezing and desiccation [9]. This likely indicates that the response of *P. murrayi* to cold shock is very similar to its response to heat shock. Therefore, *P. murrayi* may also produce heightened levels of lipids in freezing conditions.

## 5. Conclusions

The high respiratory quotient of *P. murrayi* metabolism at temperatures greater than 5 °C could portend significant changes in future MCM soil ecosystems. Without a concomitant increase in primary productivity in the MCM, such high rates of carbon turnover and CO_2_ production could potentially deplete the soil of carbon and result in an overall increase in CO_2_ production in the MCM. Alternatively, these effects may be offset by an increase of net primary productivity in lower elevation soils as new or existing phototrophs expand their range and abundance in the MCM due to ameliorated environmental conditions brought on by climate change. This would likely result in expanded abundance and distribution of *P. murrayi* in the MCM.

As temperatures in the MCM rise, so too will the metabolic activity and CO_2_ production of *P. murrayi*. To properly understand the effects of climate change on the MCM, we must first understand the response of individual soil taxa to warming temperatures. The methodology outlined in this publication is applicable to studying the effects of warming temperatures on the metabolic response of other Antarctic biota. As we further our understanding of how individual Antarctic species will respond to climate change, we will begin to gain a clearer understanding of how these changes can affect soil biodiversity and ecosystem functioning at local, regional, and perhaps even continental scales.

## Figures and Tables

**Figure 1 biology-12-00109-f001:**
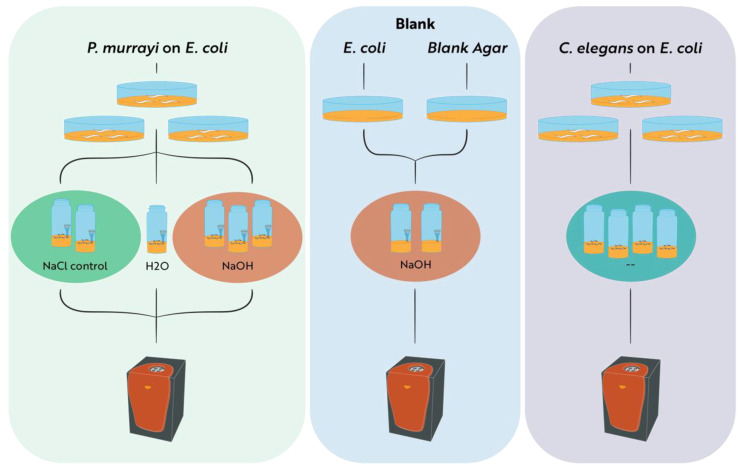
Diagram of the methodology used to measure the metabolic response of *P. murrayi* to various temperatures. Pieces of agar from viable plates were transferred to 4 mL vials and then inserted into the TAM IV. Some vials containing *P. murrayi* and *E. coli* on agar also contained a separate 250 μL ampoule with 0.4 M NaOH or 0.4 M NaCl, represented by the small cones depicted within the vials. *E. coli* and blank agar vials functioned as a negative control. Vials containing *C. elegans* were prepared for comparison without any added treatment, indicated by the “--”. The number of vials depicted is representative of the number of repetitions that were conducted. All vials were held for 4 h at each of the following temperatures sequentially: 15 °C, 10 °C, 5 °C, 15 °C, 20 °C, 25 °C, 30 °C, 35 °C, 40 °C, 45 °C, 50 °C, 15 °C. Heat rate measurements were taken every 5 s for the duration of the experiment.

**Figure 2 biology-12-00109-f002:**
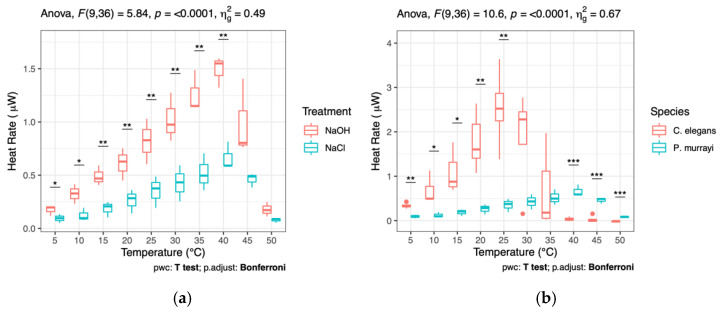
(**a**) Heat rates ($\frac{\mu J}{s}=\mu$W) per *P. murrayi* individual in vials containing NaOH are greater than those in vials without NaOH due to the reaction of NaOH with CO_2_ produced during metabolism (* *p* ≤ 0.05, ** *p* ≤ 0.01). (**b**) *C. elegans* reaches peak metabolic activity at ~25 °C, whereas *P. murrayi* experiences increasing heat rate up to 40 °C (* *p* ≤ 0.05, ** *p* ≤ 0.01, *** *p* ≤ 0.001).

**Figure 3 biology-12-00109-f003:**
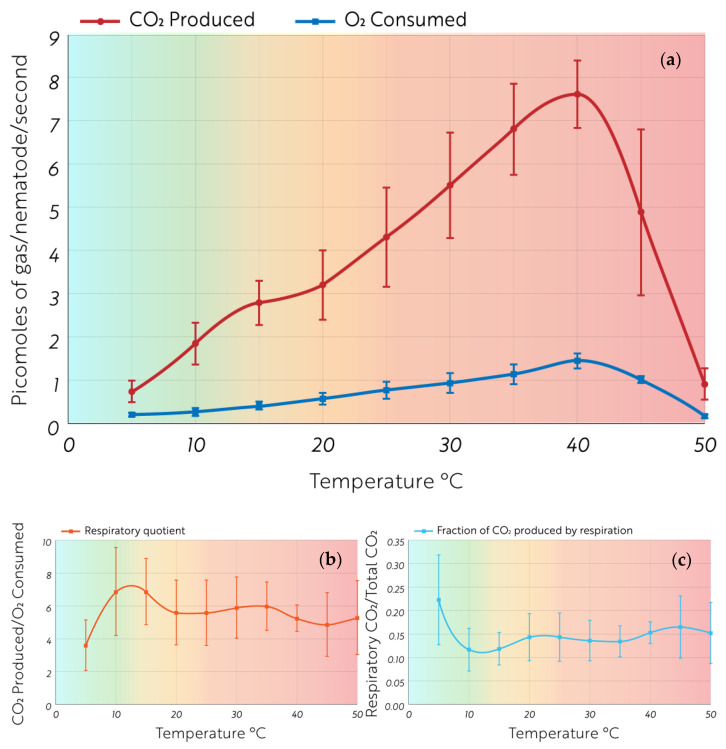
Background color represents distinct classes of summer temperatures in Taylor Valley. Blue, found between 0 and 13.02 °C, represents average summer temperatures in Taylor Valley. Yellow, found between 13.02 °C and 22.61 °C, represents the range of daily maximum temperatures in Taylor Valley. Red, found above 22.61 °C, indicates temperatures that have never been recorded in Taylor Valley. (**a**) CO_2_ production and O_2_ consumption rates as a function of temperature in *P. murrayi* individuals. (**b**) The respiratory quotient of *P. murrayi* metabolism calculated from the values in (**a**). (**c**) The fraction of CO_2_ produced by oxidative respiration at each experimental temperature.

## Data Availability

All data presented in this study are archived in the Environmental Data Initiative (EDI) Data Repository: https://doi.org/10.6073/pasta/4973e62f9a46674024e41c4f0a3769e9 (Robinson et al., 2021).

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
