# Peer review of "Temperature Response of Metabolic Activity of an Antarctic Nematode"

_biology, 2023, doi:10.3390/biology12010109_

Round 1
Reviewer 1 Report
Proposed manuscript describes an original study with an inventive design. Peak metabolic activity at 40oC of Antarctic species P. murrayi is a quite surprising result. That temperature is about 20oC higher than maximum temperature measured in Taylor Valley. Nevertheless, authors explanation about protective exaptation allowing for the recent success of P. murrayi population during the present period of warming. It will be more informative to compare metabolic capacity of P. murrayi with other Plectus species (Schulze et al. EvoDevo 2012, 3:13 http://www.evodevojournal.com/content/3/1/13).
As far as most of the results were based on extrapolation to individual nematode or 30 specimens it will be better to describe how number of nematodes in agar pieces used in experiments were counted.
Author Response
Response to Reviewer 1 Comments
Point 1: Proposed manuscript describes an original study with an inventive design. Peak metabolic activity at 40oC of Antarctic species P. murrayi is a quite surprising result. That temperature is about 20oC higher than maximum temperature measured in Taylor Valley. Nevertheless, authors explanation about protective exaptation allowing for the recent success of P. murrayi population during the present period of warming. It will be more informative to compare metabolic capacity of P. murrayi with other Plectus species (Schulze et al. EvoDevo 2012, 3:13 http://www.evodevojournal.com/content/3/1/13).
Response 1: The reviewer’s suggestion to compare P. murrayi to another Plectus species is a good one, particularly if we were seeking to uncover the evolutionary basis of this metabolic anomaly. When looking for an animal for comparative purposes, we wanted to choose another freeliving soil nematode that was not from the Antarctic, but for which the biology is well-known and genomic resources are plentiful. Thus, C. elegans seemed an optimal candidate because its biology is so well-studied and broadly understood. Other species of Plectus are less well-characterized (although thankfully, as the reviewer points out, this is changing!). For example, note that for our implication de novo lipogenesis, the metabolic pathway is only known from C. elegans (though it would not be surprising to find it across much of Nematoda). For our next steps we hope to compare transcriptomic responses to environmental insults such as this. While a draft P. murrayi genome is now readily available*, we are unaware of other Plectus genomes that are publicly available. Still, the reviewer is correct to point out that our work would be much more informative if we could place it more appropriately in a phylogenetic context. In the future we plan to do just this, but for now such activities are beyond the scope of our initial objectives, which are primarily descriptive.
Recall, our objective was to characterize the response of P. murrayi to elevated temperatures, and then use this information to inform questions about how populations of P. murrayi will respond to climate warming. Thus, having a definitive explanation for the origin and maintenance of its ecological amplitude with regard to temperature is not necessarily required for achieving the objectives of our work.
However, to address the reviewer’s concern, we note this limitation in the discussion section, stating that this impacts our ability to draw evolutionary explanations for the observed phenomena (259-261).
* Xue, X.; Suvorov, A.; Fujimoto, S.; Dilman, A.R.; Adams, B.J. Genome Analysis of Plectus Murrayi, a Nematode from Continental Antarctica. G3 Genes|Genomes|Genetics 2021, 11, doi:10.1093/g3journal/jkaa045.
* Xue, X.; Adhikari, B.N.; Ball, B.A.; Barrett, J.E.; Miao, J.; Perkes, A.; Martin, M.; Simmons, B.L.; Wall, D.H.; Adams, B.J. Ecological Stoichiometry Drives the Evolution of Soil Nematode Life History Traits. Soil Biology and Biochemistry 2023, 177, 108891, doi:10.1016/j.soilbio.2022.108891.
Point 2: As far as most of the results were based on extrapolation to individual nematode or 30 specimens it will be better to describe how number of nematodes in agar pieces used in experiments were counted.
Response 2: A phrase has been added to the methods section detailing how nematodes were counted (137-139). Exact nematode counts for each sample can be found in the dataset provided in the data availability statement.

Reviewer 2 Report
The authors conducted a laboratory study measuring metabolic capacity of an Antarctic nematode, in comparison to the highly-studied C. elegans, over increasing temperatures. They demonstrate that the Antarctic nematode, while obviously cold-adapted given its native habitat, has a very high heat tolerance. They use this finding to predict that future warming will increase P. murrayi metabolism with accompanying changes in the C cycle. The paper is overall well-written and appropriately succinct.
Overall, I find the study methods to be appropriate (though I am certainly not an expert in measuring metabolism of nematodes!) and the results to be intriguing. The finding that P. murrayi metabolism peaks at 40℃ is interesting! While we assume that cold-adapted species would therefore peak at a lower temperature that reflects their habitat, it appears that perhaps P. murrayi are maybe not so much “cold-adapted” specifically, but just super flexible and therefore tolerate both hot and cold. It’s a neat finding! My only main piece of feedback is that the eventual impacts on carbon cycling is speculative. It’s fair to be speculative in the Discussion, but with some caution. Yes, P. murrayi activity might increase with temperature, but that doesn’t mean C will become depleted unless they have no other limitations (such as water or a co-limitation by nutrients that may not be available as quickly as temperatures rise). So I suggest caution around that text in the Discussion and Conclusions, as well as the graphical abstract. In the third panel of the graphical abstract: you don’t know that’s the cause of depletion. There’s a possibility, but not something you’ve demonstrated with your data, so I recommend rewording as a potential impact. It’s currently written as fact.
Other minor feedback:
Lines 51-56: Yes, but last I read into it, this is only true of certain parts of Antarctica, not so much the McMurdo Dry Valleys. So it reads as a bit of a misdirection to use a 3℃ increase as an intro to the Dry Valleys in the next paragraph. (Unless my information is way behind…?) Also, you could maybe help segue this by pointing out whether P. murrayi lives elsewhere on the continent beyond the MCM, including in these areas where warming has been more rapid.
Line 79: It feels more accurate to me to say “could experience” rather than “is experiencing”. Just because we measure it happen in the lab doesn’t mean that’s what is happening in the ecosystem. Perhaps the rise in metabolic rate is only because they were given ample food and other resources, and those limitations would prevent a rapid rise in the field. If we can’t measure it in the field setting, it doesn’t seem that it can be stated here as though it’s happening.
Fig 3: The “2” should be a subscript on CO2 and O2.
Author Response
Response to Reviewer 2 Comments
Point 1: My only main piece of feedback is that the eventual impacts on carbon cycling is speculative. It’s fair to be speculative in the Discussion, but with some caution. Yes, P. murrayi activity might increase with temperature, but that doesn’t mean C will become depleted unless they have no other limitations (such as water or a co-limitation by nutrients that may not be available as quickly as temperatures rise). So I suggest caution around that text in the Discussion and Conclusions, as well as the graphical abstract. In the third panel of the graphical abstract: you don’t know that’s the cause of depletion. There’s a possibility, but not something you’ve demonstrated with your data, so I recommend rewording as a potential impact. It’s currently written as fact.
Response 1: The reviewer’s insight is appropriate and helpful. In the soils that P. murrayi inhabits, available organic carbon is a limiting nutrient. That, however, could change due to ameliorated environmental conditions brought on by climate change. To make this more clear we added a sentence and citation support in the discussion section (lines 290-292). We also modified some of the wording in the conclusion section to make it less speculative (340-348).
In addition, the graphical abstract has now been adjusted to include two potential outcomes: one with novel carbon inputs, and one without. These are both labeled as potential outcomes. Our study is equivocal in terms of support for either of these outcomes.
Point 2: Lines 51-56: Yes, but last I read into it, this is only true of certain parts of Antarctica, not so much the McMurdo Dry Valleys. So it reads as a bit of a misdirection to use a 3℃ increase as an intro to the Dry Valleys in the next paragraph. (Unless my information is way behind…?) Also, you could maybe help segue this by pointing out whether P. murrayi lives elsewhere on the continent beyond the MCM, including in these areas where warming has been more rapid.
Response 2: The reviewer is correct to note this. The first paragraph makes it sound as if the 3°C increase is occurring in the MCM. The MCM has, in fact, been mostly protected from temperature increases of that magnitude. However, warming is occurring in the Dry Valleys, just at a slower rate, and the frequency and magnitude of pulse warming events have increased significantly*. To address this concern and to make our statements more accurate, we have added some clarifying sentences to the introduction (55-58).
*Andriuzzi, W.S.; Adams, B.J.; Barrett, J.E.; Virginia, R.A.; Wall, D.H. Observed Trends of Soil Fauna in the Antarctic Dry Valleys: Early Signs of Shifts Predicted under Climate Change. Ecology 2018, 99, 312–321, doi:10.1002/ecy.2090.
Point 3: Line 79: It feels more accurate to me to say “could experience” rather than “is experiencing”. Just because we measure it happen in the lab doesn’t mean that’s what is happening in the ecosystem. Perhaps the rise in metabolic rate is only because they were given ample food and other resources, and those limitations would prevent a rapid rise in the field. If we can’t measure it in the field setting, it doesn’t seem that it can be stated here as though it’s happening.
Response 3: This is a very helpful criticism. We’ve made this change at line 80.
Point 4: Fig 3: The “2” should be a subscript on CO2 and O2.
Response 4: Figure three has been modified as suggested.

Reviewer 3 Report
In general, the content of this study is not 'complicated'. But the idea to study the impact of rising temperature on soil nematodes is good and the manuscript is well written.
I am not an expert in nematode physiology, but apparently, how soil organisms, especially Antarctic soil organisms, responded to climate change is important and enlightening.
I have two questions about this research:
(1) I noticed that the responses of P. murrayi and C. elegans to the temperature gradient was different, is this phenomenon exists in the same nematode populations in other regions (e.g., Plectus in USA)?
(2) soil microbial necromass is considered to be a major C sink in many soil ecosystems, so how the high C turnover rate of P. murrayi may affect the microbial necromass in Antarctic soil?
Author Response
Response to Reviewer 3 Comments
Point 1: I noticed that the responses of P. murrayi and C. elegans to the temperature gradient was different, is this phenomenon exists in the same nematode populations in other regions (e.g., Plectus in USA)?
Response 1: In this study, we simply analyzed P. murrayi and C. elegans. However, future studies could be conducted to determine whether this metabolic anomaly for P. murrayi is common across other clades of nematodes, and the Plectidiae, specifically. However, such a study lies outside the scope of our present objectives.
Point 2: Soil microbial necromass is considered to be a major C sink in many soil ecosystems, so how the high C turnover rate of P. murrayi may affect the microbial necromass in Antarctic soil?
Response 2: As P. murrayi increases in abundance and distribution across the landscape we predict that its necromass will be an important contribution to microbial metabolic processes, its carbon component ultimately exiting the soil via microbial respiration. In addition, increased rates of lipogenesis could accelerate release of CO2 via respiration, resulting in an additional net loss of carbon from the system. Whether there is a significant reduction in microbial necromass depends on rates of nematode grazing, primary productivity, and microbial turnover. None of these rates were measured in our study, however future research could focus on how temperature impacts nematode grazing rates and microbial turnover in the MCM. We have added a sentence referencing the potential for future increases in primary productivity in the Dry Valleys as a result of climate change in the discussion section (lines 318-320).

Reviewer 4 Report
The MS "Temperature Response of Metabolic Activity of an Antarctic Nematode” is an interesting article dealing with the effects of temperature on the metabolic rate of the Antarctic nematode Plectus murrayi. The work done is original and methodologically challenging, results are interesting and the discussion is comprehensible and complete. In my view, the manuscript describes an original work which produced unexpected results, which makes it even more appealing, since it opens new questions to be disentangled in the future.
My only minor recommendations deals with the discussion of the most surprising result: the high levels of CO2 produced by lipogenesis instead of respiration. I miss some more information is this regard, which I think could be at least briefly expanded. This low ratio of CO2 produced by respiration vs CO2 produced by lipogenesis (L286-288) has been reported before? Is it common? Also, I wonder if the assumption that higher temperatures would contribute to the depletion of soil C in Antarctic soils is supported by previous data: since most of the CO2 production detected in the experiment is hypothesized to come from the lipogenesis due to the ingestion of sugars in the growth medium, it is not clear if such increase would be possible in Antarctic soils. Is there any evidence of increased grazing on soil natural bacteria of P. murrayi under natural warm conditions?
Other than that, I found the MS ambitious, interesting, and well written, and deserves to be published in Biology.
Author Response
Response to Reviewer 4 Comments
Point 1: My only minor recommendations deals with the discussion of the most surprising result: the high levels of CO2 produced by lipogenesis instead of respiration. I miss some more information is this regard, which I think could be at least briefly expanded. This low ratio of CO2 produced by respiration vs CO2 produced by lipogenesis (L286-288) has been reported before? Is it common?
Response 1: In locusts, increased rates of de novo lipogenesis has been shown to cause elevated CO2 production, leading to an elevated respiratory quotient and, therefore, a smaller ratio of respiratory CO2 to total CO2. We have added a sentence referencing this along with the citation in the discussion section (lines 344-345).
As to whether this is common, it is hard to say. Very few studies have been conducted to analyze the effect of lipogenesis on respiratory quotient. However, it is likely more common in organisms that must produce high quantities of lipids for energy storage and membrane integrity. Therefore, this attribute is likely more common in extreme environments where rapid shifts in cell membrane composition and energy storage would be useful. For that reason, it might not be surprising that we see it occurring in an Antarctic ectotherm.
Point 2: Also, I wonder if the assumption that higher temperatures would contribute to the depletion of soil C in Antarctic soils is supported by previous data: since most of the CO2 production detected in the experiment is hypothesized to come from the lipogenesis due to the ingestion of sugars in the growth medium, it is not clear if such increase would be possible in Antarctic soils. Is there any evidence of increased grazing on soil natural bacteria of P. murrayi under natural warm conditions?
Response 2: This is a good point. We have not measured the effect of temperature on P. murrayi grazing in natural warm conditions. This would be a good topic for future research. What we see in our study is that, as energy increases, more carbon is utilized in lipogenesis. Whether this would occur in the natural habitat would likely rely on organic soil carbon availability. We have added a sentence addressing this limitation in the discussion section (295-297). Carbon availability will likely decrease as P. murrayi metabolic activity increases, as soil organic carbon is a limiting nutrient in MCM soils. However, climate change may lead to increased primary productivity in the Dry Valleys, resulting in increased organic soil carbon. To address this, we have added a sentence mentioning this along with a citation in the discussion section (318-320).
